# Antimicrobial Resistance of *Escherichia coli* for Uncomplicated Cystitis: Korean Antimicrobial Resistance Monitoring System

**DOI:** 10.3390/antibiotics13111075

**Published:** 2024-11-12

**Authors:** Seong Hyeon Yu, Seung Il Jung, Seung-Ju Lee, Mi-Mi Oh, Jin Bong Choi, Chang Il Choi, Yeon Joo Kim, Dong Jin Park, Sangrak Bae, Seung Ki Min

**Affiliations:** 1Department of Urology, Chonnam National University Hospital, Chonnam National University Medical School, Gwangju 61469, Republic of Korea; domer12@hanmail.net; 2Department of Urology, The Catholic University of Korea, St. Vincent’s Hospital, Suwon 16247, Republic of Korea; lee.seungju@gmail.com; 3Department of Urology, Korea University Guro Hospital, Seoul 08308, Republic of Korea; mamah77@paran.com; 4Department of Urology, Bucheon St. Mary’s Hospital, College of Medicine, The Catholic University of Korea, Seoul 14647, Republic of Korea; c-sparrow@hanmail.net; 5Department of Urology, Hallym University Dongtan Sacred Heart Hospital, College of Medicine, Hallym University, Hwaseong 18450, Republic of Korea; choicog@gmail.com; 6Department of Urology, Daegu Fatima Hospital, Daegu 41199, Republic of Korea; winewiner85@gmail.com; 7Department of Urology, Dongguk University College of Medicine, Gyeongju 38066, Republic of Korea; parkdj0510@gmail.com; 8Department of Urology, Uijeongbu St. Mary’s Hospital, College of Medicine, The Catholic University of Korea, Seoul 11765, Republic of Korea; robinbae97@catholic.ac.kr; 9Department of Urology, Goldman Urologic Clinic, Seoul 05510, Republic of Korea; msk0701@hanmail.net

**Keywords:** urinary tract infection, cystitis, *Escherichia coli*, antibiotic resistance

## Abstract

**Objectives:** Uncomplicated cystitis is a leading form of bacterial UTI; the most common causative bacterium worldwide is *Escherichia coli*. This internet-based, prospective, multicenter, and national observational study aimed to report the antimicrobial resistance of *E. coli* in patients with uncomplicated cystitis through the use of the Korean Antimicrobial Resistance Monitoring System (KARMS) in 2023. **Results:** Data for a total of 654 patients were retrieved from the KARMS database. The mean (standard deviation) patient age was 55.9 (18.3) years. The numbers of postmenopausal women and patients with recurrent cystitis were 381 (59.4%) and 78 (11.9%), respectively. Regarding antimicrobial susceptibility, 96.8% were susceptible to fosfomycin, 98.9% to nitrofurantoin, 50.9% to ciprofloxacin, and 82.4% to cefotaxime. Extended-spectrum beta-lactamase positivity was 14.4% (89/616), and was significantly higher in tertiary hospitals (24.6%, *p* < 0.001) and recurrent cystitis (27.6%, *p* < 0.001). Fluoroquinolone resistance was significantly higher in tertiary hospitals (57.8%, *p* < 0.001), postmenopausal women (54.2%, *p* < 0.001), and recurrent cystitis (70.3%, *p* < 0.001). In addition, postmenopausal status (95% confidence interval [CI]: 1.44–3.17, odds ratio [OR] 2.13, *p* < 0.001), recurrent cystitis (95% CI: 1.40–4.66, OR 2.56, *p* = 0.002) and tertiary hospitals (95% CI: 1.00–2.93, OR 1.71, *p* = 0.049) were associated with significantly increased fluoroquinolone resistance. **Methods:** Any female patient diagnosed with clinical uncomplicated cystitis and microbiologically proven *E. coli* infection in 2023 was eligible for this study. Patient data were obtained from the web-based KARMS database. The antimicrobial susceptibility of *E. coli* was analyzed according to clinical factors, including hospital region, hospital type, menopause status, and recurrence status. **Conclusions:** The antimicrobial resistance of *E. coli* in patients with uncomplicated cystitis in the Republic of Korea has reached a serious level, especially in fluoroquinolone resistance. Therefore, major efforts should be made to reduce antimicrobial resistance.

## 1. Introduction

Urinary tract infection (UTI) is among the most common bacterial infections in women, affecting one million people worldwide each year, and is classified as uncomplicated or complicated based on the clinical presentation, the anatomical level, and the severity of infection [1,2]. Uncomplicated cystitis (UC) accounts for the greatest number of UTIs, typically affecting healthy non-pregnant women with no anatomic or functional urinary tract abnormalities [3,4]. Almost half of all women experience at least one episode of UC during their lifetime, mainly accompanied by symptoms such as dysuria, urgency, frequency, and a feeling of incomplete bladder emptying [5,6]. Of note, UC is not only common but also has a high recurrence rate; nearly 30% of women who have a UC will have a recurrence within 6–12 months despite treatment [7]. The aforementioned populations are particularly vulnerable to recurrent UC, which can lead to a socioeconomic burden with a negative impact on factors such as the quality of relationships, self-esteem, and the capacity for work [8].

UC can be caused by a variety of pathogens, such as Gram-negative and Gram-positive bacteria as well as fungi [9]. Uropathogenic *Escherichia coli*, a heterogeneous group of extraintestinal pathogenic *E. coli*, is the most common causative bacterial pathogen for UC worldwide, accounting for 75–90% of cases [1,10,11]. Therefore, clinicians have traditionally treated UC with empirical antimicrobials, and indeed, the use of antimicrobials has been the most important approach to the treatment of UC, significantly reducing complications and improving quality of life [12]. However, the long-term and injudicious use of antimicrobial agents has promoted the gradual alteration of the normal microbiota and the development of antimicrobial resistance (AMR) [13]. Consequently, the emergence of antimicrobial-resistant bacteria, such as fluoroquinolone-resistant *E. coli* and extended-spectrum ß-lactamase (ESBL)-producing *E. coli*, has proved challenging for clinicians treating UTIs, as these bacteria are no longer responsive to antimicrobial agents and can lead to serious complications, including severe illness, prolonged hospitalization, and death [14]. Moreover, in recent years, the proportion of antimicrobial-resistant bacteria, especially multidrug-resistant bacteria (e.g., carbapenem-resistant Enterobacterales), has been rapidly increasing worldwide, and is now considered one of the greatest threats to global health [11,15,16,17]. In fact, numerous organizations (e.g., the World Health Organization and Centers for Disease Control and Prevention) have declared that AMR is a ‘global public health problem’ [18,19]. Thus, the appropriate use of antimicrobial agents is a cornerstone of treating UTIs, and should be based on regional microbiologic characteristics and antibiotic susceptibility information. The investigation and monitoring of AMR is imperative to obtain the correct information on the distribution, microbiologic characteristics, and antibiotic susceptibility of uropathogens in the community.

To date, there have been several efforts to investigate and monitor AMR in the Republic of Korea [20,21,22]. Recently, the Korean Association of Urogenital Tract Infection and Inflammation (KAUTII) established a monitoring system for AMR surveillance, named the Korean Antimicrobial Resistance Monitoring System (KARMS). The KARMS for UTIs is an internet-based study (https://uti.karms.kr/main-page, accessed from 1 January 2023), and is ongoing with the nationwide cooperation of medical centers in the Republic of Korea. The present study aimed to report AMR focusing on the UC patients with microbiologically proven *E. coli*, the most common causative bacterium for UC worldwide, in the KARMS database in 2023.

## 2. Results

### 2.1. Baseline Characteristics

A total of 885 patients with UC were included in the KARMS database, and 654 patients (75.1%) with microbiologically proven *E. coli* were eligible for the present study (Table 1). The other causative bacterial pathogens for UC were *Klebsiella pneumoniae* (6.2%, 54), followed by *Enterococcus* spp. (4.1%, 36), *Streptococcus* spp. (3.8%, 33), and *Proteus mirabilis* (3.0%, 26).

The mean (standard deviation) patient age was 55.87 (18.31) years. The numbers of patients in primary, secondary, and tertiary medical centers were 271 (30.7%), 418 (48.0%) and 195 (34.0%), respectively. Regarding hospital region, most patients were in Gyeonggi Province (*n* = 264, 40.4%), followed by Seoul (189, 28.9%), Jeolla (99, 15.1%), Gyeongsang (78, 11.9%), and Chungcheong (24, 3.7%). The numbers of patients with community-acquired UTIs, postmenopausal status, and recurrent cystitis were 648 (99.2%), 260 (40.6%), and 78 (11.9%), respectively. ESBL positivity was 14.4% (89/616).

### 2.2. Antimicrobial Susceptibility of E. coli

Overall, 50.9% (324/636) were susceptible to ciprofloxacin, 82.4% (521/632) to cefotaxime, 96.8% (61/63) to fosfomycin, 98.9% (187/189) to nitrofurantoin, 99.7% (627/629) to amikacin, 82.7% (497/601) to amoxicillin/clavulanate, 61.0% (108/177) to levofloxacin, 68.1% (399/586) to trimethoprim/sulfamethoxazole, 97.5% (624/640) to piperacillin/tazobactam, and 100% (718/718) to ertapenem (Table 2). Thus, the resistance rates for *E. coli* to ciprofloxacin and cefotaxime in patients with UC were 49.1% and 17.6%, respectively.

### 2.3. Assessment of E. coli AMR by Clinical Factors

The rates of ESBL positivity, fluoroquinolone resistance, and third-generation cephalosporin resistance were compared according to clinical factors such as hospital type, hospital region, menopausal status, and recurrence. There were no significant differences observed according to hospital region. ESBL positivity was significantly higher in tertiary hospitals (primary 7.8%, secondary 13.8%, tertiary 24.6%; *p* < 0.001) and patients with recurrent cystitis (non-recurrence 12.4%, recurrence 27.6%; *p* < 0.001). Fluoroquinolone resistance was significantly higher in tertiary hospitals (primary 31.7%, secondary 47.1%, tertiary 57.8%; *p* < 0.001), postmenopausal women (premenopausal 30.6%, postmenopausal 54.2%; *p* < 0.001), and patients with recurrent cystitis (non-recurrence 41.0%, recurrence 70.3%; *p* < 0.001). In addition, resistance to third-generation cephalosporins was significantly higher in tertiary hospitals (primary 11.3%, secondary 17.1%, tertiary 23.0%; *p* = 0.033) and postmenopausal women (premenopausal 12.6%, postmenopausal 19.6%; *p* = 0.004) (Figure 1).

### 2.4. Assessment of Clinical Factors Associated with Fluoroquinolone Resistance and ESBL Positivity

Tertiary hospitals (95% confidence interval [CI]: 1.00–2.93, odds ratio [OR] 1.71; *p* = 0.049), postmenopausal status (95% CI: 1.44–3.17, OR 2.13; *p* < 0.001), and recurrent cystitis (95% CI: 1.40–4.66, OR 2.56; *p* = 0.002) were associated with increased fluoroquinolone resistance (Table 3). In addition, tertiary hospitals (95% CI: 1.59–6.26, OR 3.15; *p* = 0.001) and recurrent cystitis (95% CI: 1.01–3.41, OR 1.86; *p* = 0.045) were associated with increased ESBL positivity (Table 4).

## 3. Discussion

UC is a leading form of bacterial UTI, affecting healthy women with no structural or neurologic urinary tract abnormalities, and can be accompanied by various symptoms, such as frequency, urgency, and dysuria, and has a socioeconomic burden due to frequent recurrences [1,3,4]. In general, UC due to *E. coli* is simple to treat with oral empiric antibiotics. However, AMR has been rapidly increasing globally [11,15,16,17]; therefore, investigation and monitoring of AMR are essential to select the optimal antibiotics based on the correct antimicrobial susceptibility information in the community. The present study was conducted focusing on the AMR of *E. coli* in patients with UC, and the results were that fluoroquinolone resistance was 49.1%, and was significantly higher in tertiary hospitals, postmenopausal women, and patients with recurrent cystitis.

Uropathogenic *E. coli*, a Gram-negative bacillus of the Enterobacterales family, commonly originates from the human gastrointestinal tract and various animals [23,24]. *E. coli* is the most common pathogenic factor for UTIs, and the prevalence was reported as 70–75% in both adult and pediatric UTIs in previous studies worldwide, and in the present study [9,21,25,26]. The pathogenic process of *E. coli* in the urinary tract can be explained by several steps: periurethral and vaginal invasion and colonization, ascension into the bladder lumen and growth in the urine, adherence to the surface and interaction with the defensive system of the bladder epithelium, biofilm formation, replication via the formation of intracellular bacterial communities (a source of quiescent intracellular reservoirs), and kidney colonization [9,24]. This process can cause severe sequalae such as recurrent UTIs, bacteremia, septicemia, urosepsis, and even death, despite treatment with empiric antibiotics [27,28]. In fact, at least 30% of patients with UC have a recurrence within 6–12 months, leading to a socioeconomic burden from factors such as a negative impact on the quality of relationships, self-esteem, and capacity for work [8].

The use of antimicrobial agents is the most important approach to the treatment of UTIs, and has significantly reduced mortality and increased life expectancy [12]. However, the persistent failure to develop new antimicrobial agents and the injudicious use of antimicrobials have led to the emergence of AMR [13]. Moreover, the long-term use of antimicrobial agents can promote the gradual alteration of the normal microbiota of the vagina and gastrointestinal tract and the development of multidrug-resistant microorganisms [29]. Indeed, the AMR of bacteria, such as ESBL-producing *E. coli* and multidrug-resistant bacteria, has been rapidly increasing in recent years and has become a major concern worldwide [11,15,16,17]. Numerous organizations such as the World Economic Forum, the World Health Organization, the Centers for Disease Control and Prevention, and the Infectious Diseases Society of America have declared that AMR is a ‘global public health problem’ [18,19]. In addition, a report on the socioeconomic burden of AMR predicts that, in 2050, about 444 million people will experience infections, at a global economic cost of approximately USD 120 trillion [15]. Hence, the appropriate selection and use of optimal antibiotic therapies is imperative to reduce AMR and its effect on socioeconomic burden, and the systemic surveillance of AMR is also crucial. To date, there have been several efforts to improve AMR surveillance: for example, the Global Prevalence of Infections in Urology study from 2005 was a representative, web-based study [20]. In the Republic of Korea, there was a nationwide survey in 2008 for the AMR of urinary isolates in UTIs [21]; this survey has become a good model of the current AMR monitoring system established by KAUTII in 2023. KARMS is an internet-based surveillance system, with input from medical centers nationwide in Korea, and has the advantage that it can be conducted continuously with reduced time and cost. In practice, most major urologic centers in Korea participate in KARMS, so that data can be obtained on regional microbiologic characteristics and corresponding antibiotic susceptibility information. From the KARMS database, the present study was conducted, focusing on the AMR for *E. coli* in patients with UC.

In the present study, the resistance rates for *E. coli* to ciprofloxacin and cefotaxime were 49.1% and 17.6%, respectively. These rates were higher than those in a South Korean nationwide survey in 2008: 25% resistance to fluoroquinolones, and 5% resistance to third-generation cephalosporins [21]. Distressingly, in just 15 years, resistance rates to fluoroquinolones and third-generation cephalosporins have approximately doubled and tripled, respectively. In 2008, the prevalence of *E. coli* was reported as 73%, which is similar to the prevalence of 75% in the present study.

Results from other countries showed that resistance rates for *E. coli* to fluoroquinolones were significantly higher in developing countries (Nepal 64.6%, Pakistan 60.8%, Mongolia 58.1%, Jordan 55.5%, Ethiopia 85.5%) than in developed countries (USA 5.1%, Germany 10.5%, Switzerland 17.4%, France 24.8%, Japan 18.6%) [22,24]. Compared to other countries, the current fluoroquinolone resistance rate for *E. coli* in patients with UC in the Republic of Korea has already reached serious levels. Hence, fluoroquinolones should no longer be used as empiric antibiotics for UC. However, fosfomycin and nitrofurantoin can be proposed as reliable empiric antibiotics for UC, based on the low resistance rates to fosfomycin and nitrofurantoin in the present study; this recommendation is consistent with recent studies and guideline recommendations for the clinical and microbiologic treatment of UC [30].

In the present study, AMR rates for *E. coli* to fluoroquinolones and third-generation cephalosporins were evaluated according to clinical factors, such as hospital region and type, menopausal status, and recurrence status. Regarding hospital region, no significant differences were observed. However, regarding other clinical factors, fluoroquinolone resistance was significantly higher in tertiary hospitals (primary 31.7%, secondary 47.1%, tertiary 57.8%), postmenopausal women (premenopausal 30.6%, postmenopausal 54.2%), and patients with recurrent cystitis (non-recurrence 41.0%, recurrence 70.3%; *p* < 0.001); all these factors were associated with increased fluoroquinolone resistance. Of note, fluoroquinolone resistance in patients with recurrent cystitis was extremely high, at 70.3%, which means that fluoroquinolones should no longer be used in these patients. In addition, resistance to third-generation cephalosporins was significantly higher in tertiary hospitals (primary 11.3%, secondary 17.1%, tertiary 23.0%) and postmenopausal women (premenopausal 12.6%, postmenopausal 19.6%). These significant differences have several explanations. For example, lower levels of estrogen at menopause lead to adverse changes in the urogenital epithelium and urogenital microbiome by reducing urothelial thickness and an abundance of tight-junction proteins, which can promote pathogen colonization via impaired urothelial defense mechanisms [31]. Multiple and long-term use of antibiotics in patients with recurrent cystitis increases AMR via dysbiosis induced by the breakdown of microbiota homeostasis and quiescent intracellular reservoirs formed by infected intracellular bacterial communities [29]. Also, many patients in tertiary hospitals, compared to other types of hospitals, are postmenopausal women and patients with recurrent cystitis.

In a detailed analysis, the rates of fluoroquinolone resistance in premenopausal and postmenopausal women in the Republic of Korea were 30.6% and 54.2%, respectively. Compared to the results of a recent Japanese study, [22] the South Korean results showed 3-fold higher rates in premenopausal women and 22% higher rates in postmenopausal women. Even considering the differences in the representative types of fluoroquinolones used in the two countries (Korea: ciprofloxacin, Japan: levofloxacin), the present study showed that fluoroquinolone resistance in Korea is incredibly high. Conversely, resistance rates to third-generation cephalosporins in premenopausal and postmenopausal women in Korea were 12.6% and 19.6%, respectively, and there were no significant differences compared to the recent Japanese study [22].

ESBL-producing Gram-negative bacteria have gradually increased in prevalence and have now become endemic worldwide, in part as a consequence of the use, abuse, and misuse of β-lactam antibiotics. In general, ESBL-producing *E. coli* are resistant to most β-lactam antibiotics, including extended-spectrum cephalosporins and monobactams, and are commonly identified in both community-acquired infections and healthcare-associated infections [32]. The prevalence of ESBL-producing *E. coli* has been reported as 8–30%, according to geographic location, and their prevalence in the Asia–Pacific region has been reported as 17.4% [11]. In the present study, ESBL positivity was 14.4%, and was significantly higher in tertiary hospitals (primary 7.8%, secondary 13.8%, tertiary 24.6%) and patients with recurrent cystitis (non-recurrence 12.4%, recurrence 27.6%); both factors were associated with increased ESBL positivity. These significant differences can be explained by the distribution of patients according to hospital type, and age- or infection-related hormonal, mechanical, and physiological alterations [29,31].

The results of the present study indicate that the AMR of *E. coli* in patients with UC in the Republic of Korea has reached a serious level, especially regarding fluoroquinolone resistance. Although there are many contributing factors to increasing AMR, probably the most important are the injudicious use of antibiotics, and nonadherence to local, national, or international guidelines. In a survey of actual treatment practices for patients with UC in Korea, 28% of urologists had not performed a urine culture at initial treatment, and 54% had not performed a urine culture, even in cases of initial treatment failure. In addition, the most commonly prescribed antibiotics were second- or third-generation cephalosporins and fluoroquinolones, contrary to guideline recommendations for the use of fosfomycin or nitrofurantoin [4,33].

In the present study, the rate of resistance to third-generation cephalosporins was high. If this trend continues, the rate of resistance to third-generation cephalosporins will eventually reach a serious level, like the rate of resistance to fluoroquinolones. Hence, antimicrobial stewardship is required to reduce AMR. The goal of antimicrobial stewardship is to optimize clinical outcomes and ensure cost-effective therapy while minimizing unintended consequences of antimicrobial use [34]. Antimicrobial stewardship consists of a few programs: regular training of staff in the best use of antimicrobial agents; adherence to local, national, or international guidelines; audits of adherence and treatment outcomes; regular monitoring and feedback to prescribers of their performance; and local pathogen resistance [35]. Moreover, antimicrobial stewardship promotes increased adherence to antibiotic policy, leading to reduced antibiotic treatment duration and a reduced relative risk of mortality [35]. Therefore, major efforts should be made to establish antimicrobial stewardship along with AMR surveillance in the Republic of Korea.

The present study has some limitations. First, although this was an internet-based, prospective, multicenter, and national observational survey, there may have been a selection bias due to a variable number of patients included by hospital region and type. Second, the methodology used to identify *E. coli* and antimicrobial susceptibility was not unified. However, all tests performed at participating institutions were approved by the microbial standards of each institution, and antimicrobial susceptibility was investigated according to the recommendations of the Clinical and Laboratory Standards Institute (CLSI) [36,37,38]. Third, data entry into the website may have been influenced by inter-investigator bias.

Nevertheless, the present study has some possibilities and strengths. In circumstances where AMR has emerged as a one of the greatest threats to global health, the present study provides important information on the AMR status of *E. coli* in patients with UC in the Republic of Korea, which can help clinicians to understand the severity of AMR and avoid the injudicious use of antibiotics. Also, the continuous monitoring and accumulation of data enables the identification of changing AMR patterns over time and comparisons with other antimicrobial monitoring systems. In addition, our data can provide the evidence base for future antimicrobial treatment guidelines for UTIs.

## 4. Materials and Methods

### 4.1. KARMS and Participating Institutions

KARMS was established in 2023 for the surveillance of AMR in patients with UTIs, and is an internet-based, prospective, multicenter, and national observational survey. KARMS is endorsed by KAUTII and carried out through the website (https://uti.karms.kr/main-page, accessed from 1 January 2023) with the cooperation of medical centers throughout Korea. In 2023, a total of 31 hospitals (including 8 private clinics) participated in KARMS. The participating hospitals were in five administrative provinces (Seoul, Gyeonggi, Chungcheong, Jeolla, and Gyeongsang).

### 4.2. Data Collection and Study Design

Thirty-five urologists participated in the present study by inputting data for patients with UTIs into the website. Nationality, age, sex, outpatient or inpatient, association (community-acquired, healthcare-associated, hospital-acquired), uncomplicated or complicated, classification of UTI (cystitis, pyelonephritis, catheter-associated UTI, acute prostatitis, chronic prostatitis, urosepsis, or asymptomatic bacteriuria), and recurrence or non-recurrence were investigated. For women, menopausal status was also investigated. Uncomplicated cystitis was defined as acute, sporadic, or recurrent cystitis limited to non-pregnant women with no known relevant anatomic and functional abnormalities within the urinary tract or comorbidities [4,5]. In addition, the clinical and laboratory criteria for patients with UC were defined as (1) the presence of microbes by urine culture (>10^3^ colony forming units/mL); (2) clinical symptoms such as dysuria, urgency, frequency, and feeling of incomplete bladder emptying; and (3) results of urinalysis (white blood cell count >5 cells/high power field). Recurrent UTIs were defined as recurrences of uncomplicated and/or complicated UTIs with a frequency of at least three UTIs per year or two UTIs in the last 6 months [8].

Among the entire UTI data from the KARMS database, any female patient aged ≥18 years diagnosed with clinical UC and microbiologically proven *E. coli* between January 2023 and December 2023 was included in the present study. Hence, the data for patients with UC and microbiologically proven *E. coli* between January 2023 and December 2023 were retrieved and analyzed. Patient with age < 18 years, mixed or negative growths in urine culture, complicated cystitis, and other UTIs (pyelonephritis, catheter-associated UTI, prostatitis, urosepsis, or asymptomatic bacteriuria) were excluded from the present study.

### 4.3. Antimicrobial Susceptibility Testing

All patients in the present study had a urine culture and isolated uropathogenic *E. coli*. ESBL positivity and antimicrobial susceptibility tests were performed in the local laboratories according to the microbial standards of each institute. In detail, the VITEK^®^ 2 (bioMérieux, Inc., Durham, NC, USA) system was used to perform antimicrobial susceptibility and confirmation assays. For the Vitek 2 system, briefly, a selected colony from the sample was subcultured onto a nutrient medium to produce a fresh colony. After an 18–24 h incubation, this colony was mixed into 3.0 mL of sterile saline (0.45% NaCl), with turbidity adjusted to match the 0.5~0.63 McFarland standard. The prepared suspension, along with an AST card, was then placed in the cassette holder to initiate automated susceptibility testing. Minimum inhibitory concentrations (MICs) of each antibiotic were measured by the agar dilution method according to the recommendations of the CLSI [36]. Antimicrobial susceptibility was interpreted as S (susceptible), I (intermediate), or R (resistant), according to the breakpoints by CLSI recommendations [37,38]. The results of the antimicrobial susceptibility testing were obtained according to each antimicrobial agent: amikacin, amoxicillin/clavulanate, cefazolin, cefepime, cefotaxime, cefoxitin, ceftazidime, ciprofloxacin, ertapenem, fosfomycin, gentamycin, imipenem, levofloxacin, meropenem, nitrofurantoin, piperacillin/tazobactam, and trimethoprim/sulfamethoxazole.

### 4.4. Statistical Analyses

Statistical analyses were performed using STATA version 16.1 software (StataCorp; College Station, TX, USA). Descriptive analyses were performed to assess patient demographics. Continuous variables are presented as means and standard deviations, and categoric variables are presented as frequencies (%). ESBL positivity, fluoroquinolone resistance, and resistance to third-generation cephalosporins were compared according to clinical factors (hospital region, hospital type, menopause status, and recurrence of infection) using Pearson’s two-sided test and Fisher’s exact test. A logistic regression test was performed to identify clinical factors associated with ESBL positivity and fluoroquinolone resistance. A *p* value < 0.05 was considered statistically significant.

### 4.5. Ethics Statement

The study protocol was reviewed and approved by the institutional review board of Chonnam National University Hospital (approval number CNUH-2024-085). This study was performed in accordance with the principles of the Declaration of Helsinki and the Ethical Guidelines for Clinical Studies.

## 5. Conclusions

AMR has been rapidly increasing globally. Therefore, investigation and monitoring of AMR are indispensable to select the optimal antibiotics based on the correct antimicrobial susceptibility information. In this context, the present study provides important information on the AMR status of *E. coli* in patients with UC and showed that fluoroquinolone resistance in the Republic of Korea has reached a serious level, and was significantly higher in tertiary hospitals, postmenopausal women, and patients with recurrent cystitis. In addition, the present study will be a useful future reference for continuous surveillance over the years. Finally, considering the current situation for increasing AMR, major efforts such as continuous monitoring, data accumulation, and appropriate antimicrobial stewardship should be required to reduce AMR in the future.

## Figures and Tables

**Figure 1 antibiotics-13-01075-f001:**
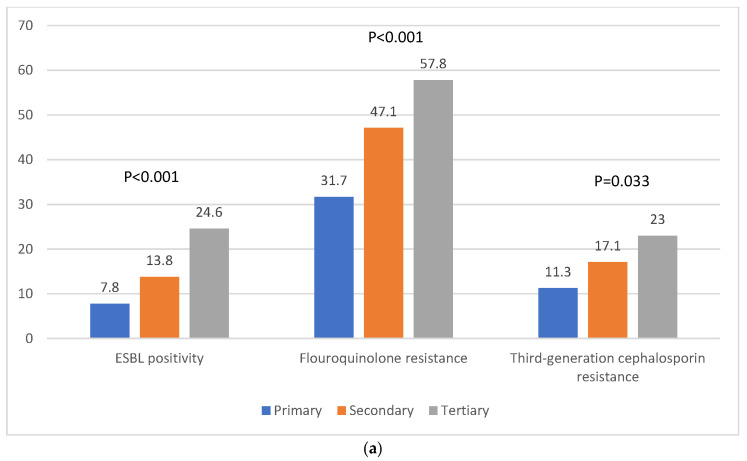
ESBL positivity, fluoroquinolone resistance, and third-generation cephalosporin resistance of *E. coli* according to clinical factors. (**a**) Hospital type, (**b**) menopausal status, and (**c**) recurrence of infection.

**Table 1 antibiotics-13-01075-t001:** Demographic characteristics of patients with uncomplicated cystitis and microbiologically proven *E. coli* infection.

Variables ^a^	N = 654
Age (years)	55.9 ± 18.3
Hospital type	
Primary	271 (30.7%)
Secondary	418 (47.3%)
Tertiary	195 (22.1%)
Hospital region	
Seoul	189 (28.9%)
Gyeonggi	264 (40.4%)
Gyeongsang	78 (11.9%)
Jeolla	99 (15.1%)
Chungcheong	24 (3.7%)
Postmenopausal	260 (40.6%)
UTI classification	
Community-acquired	648 (99.2%)
Healthcare-associated	3 (0.5%)
Hospital-acquired	2 (0.3%)
Recurrent cystitis	78 (11.9%)
ESBL-positive (*n* = 616)	89 (14.4%)

^a^ Data are shown as mean ± standard deviation for continuous variables and number (%) for categoric variables. ESBL, extended-spectrum ß-lactamase; UTI, urinary tract infection.

**Table 2 antibiotics-13-01075-t002:** Antimicrobial susceptibility of *E. coli* in patients with uncomplicated cystitis.

Antibiotic Susceptibility	Susceptible Number/Investigated Number
Amikacin	627/629 (99.7%)
Amoxicillin/clavulanate	497/601 (82.7%)
Cefazolin	394/538 (73.2%)
Cefepime	576/646 (89.2%)
Cefotaxime	521/632 (82.4%)
Cefoxitin	432/469 (92.1%)
Ceftazidime	549/622 (88.3%)
Ciprofloxacin	324/636 (50.9%)
Levofloxacin	108/177 (61.0%)
Ertapenem	593/593 (100.0%)
Fosfomycin	61/63 (96.8%)
Gentamicin	500/643 (77.8%)
Imipenem	640/640 (100.0%)
Meropenem	265/266 (99.6%)
Nitrofurantoin	187/189 (98.9%)
Piperacillin/tazobactam	624/640 (97.5%)
Trimethoprim/sulfamethoxazole	399/586 (68.1%)

Data are shown as number (%) for categoric variables.

**Table 3 antibiotics-13-01075-t003:** Demographic and clinical factors associated with fluoroquinolone resistance.

Variables	Univariate Analysis		Multivariate Analysis	
	Odds Ratio (95% CI)	*p* Value	Odds Ratio (95% CI)	*p* Value
Hospital region				
Seoul	Reference			
Gyeonggi	0.85 (0.57–1.25)	0.399		
Gyeongsang	1.23 (0.71–2.10)	0.461		
Jeolla	1.33 (0.79–2.22)	0.278		
Chungcheong	0.65 (0.26–1.62)	0.351		
Hospital type				
Primary	Reference		Reference	
Secondary	1.89 (1.29–2.76)	0.001	1.23 (0.80–1.90)	0.347
Tertiary	3.23 (2.02–5.14)	<0.001	1.71 (1.00–2.93)	0.049
Postmenopausal	2.69 (1.91–3.80)	<0.001	2.13 (1.44–3.17)	<0.001
Recurrent cystitis	3.62 (2.08–6.29)	<0.001	2.56 (1.40–4.66)	0.002

CI, confidence interval.

**Table 4 antibiotics-13-01075-t004:** Demographic and clinical factors associated with ESBL positivity.

Variables	Univariate Analysis		Multivariate Analysis	
	Odds Ratio (95% CI)	*p* Value	Odds Ratio (95% CI)	*p* Value
Hospital region				
Seoul	Reference			
Gyeonggi	1.22 (0.71–2.10)	0.469		
Gyeongsang	0.51 (0.20–1.31)	0.163		
Jeolla	0.92 (0.45–1.89)	0.818		
Chungcheong	1.60 (0.55–4.67)	0.390		
Hospital type				
Primary	Reference		Reference	
Secondary	1.88 (1.01–3.52)	0.048	1.75 (0.93–3.29)	0.083
Tertiary	3.84 (2.00–7.36)	<0.001	3.15 (1.59–6.26)	0.001
Postmenopausal	1.56 (0.96–2.54)	0.072		
Recurrent cystitis	2.69 (1.53–4.73)	0.001	1.86 (1.01–3.41)	0.045

ESBL, extended-spectrum β-lactamase.

## Data Availability

The data presented in this study are available in this article.

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
