# Peer review of "Antimicrobial Resistance of Escherichia coli for Uncomplicated Cystitis: Korean Antimicrobial Resistance Monitoring System"

_antibiotics, 2024, doi:10.3390/antibiotics13111075_

Round 1
Reviewer 1 Report
Comments and Suggestions for Authors
Based on a web-based survey among urologists, the authors present a country-wide report on uncomplicated cystitis caused by E. coli and its susceptibility in Korea. These studies are regionally necessary to determine empirical antibiotic treatments and monitor antimicrobial resistance. However, some issues need to be clarified.
General comments:
The authors should clarify in the text that the study included only women, although uncomplicated cystitis is more frequent in women; it should be clarified in the abstract and the main text
The authors should discuss why only E. coli isolates were included in the survey, or at least comment on the proportion of UC caused by other pathogens, or clarify if the web page only allows E. coli isolates to be reported.
It appears to be a mismatch or a definition confusion about the cefotaxime-resistant isolates and the ESBL isolates; in fact, the criteria for determining an isolate as ESBL was not mentioned (e.g., phenotypic tests, Vitek, molecular methods) or if these isolates we’re only defined by the third generation cephalosporins non-susceptibility (which may be inappropriate). The authors should clarify this.
For the use of the term uropathogenic E. coli, it is unclear if any molecular testing was conducted on uropathogenic E. coli or if the isolates are considered to be such because of the isolation site.
Despite the previous use of antibiotics being mentioned as one of the variables collected, no results are shown. The authors should clarify this.
Laboratory methods for bacterial identification and susceptibility testing should be stated, even if they are not uniform among the centers. It should be mentioned which proportion of the centers used automated methods for susceptibility and identification. If available, MIC 50 and MIC 90 for each antibiotic should be noted.
Despite the authors report 654 isolates/patients, it seems to be a small number for a multicenter study; authors should discuss how representative of the country the included sample is (national cases reported in one year, other epidemiological reports)
Other comments:
Line 27: “The mean (standard deviation) patient age was 55.87 (18.31) years”
Comment: Authors may consider removing one decimal from the number
Line 291:4.2. Data Collection and Study Design
Comment: The dates of enrollment should be clearly defined since it is not clear if data was acquired from January to December 2023 or on other dates. Also, inclusion and exclusion criteria should be stated.
Author Response
All authors sincerely thank you for your kind comments of the manuscript. Please refer to attached file.

Reviewer 2 Report
Comments and Suggestions for Authors
I have read with interest the manuscript submitted by Yu et al., since AMR represents a global concern.
I have a few comments to be addressed in order to improve the quality of the manuscript:
- the abstract could benefit from an introduction, a phrase with some general information.
- row 44 - not only bacterial, although this is the main etiology.
- the introduction should be expanded
- replace Enterobacteriaceae with Enterobacterales
- in the results section, consider merging some figures, for a better visual aspect of the manuscript and understanding
- The novelty element in this manuscript is quite low. I suggest proposing some more specific recommendations for this issue, to enhance the manuscript's strenghtness.
- the reference list is scarce, as the literature on this topic is vast.
Author Response

(The authors gave the same response as above.)

Reviewer 3 Report
Comments and Suggestions for Authors
Please find below my comments on the manuscript:
-Abstract, the type of study is missing please add.
-Methods: the type of study is also missing and should be added.
-Results Table 3, Please change the table title: “Clinical factors associated with fluoroquinolone resistance.” to “Demographic and clinical factors associated with fluoroquinolone resistance” as the factor included in the table such as hospital region is not a clinical one but more a demographic one. This is also for Table 4.
Author Response

(The authors gave the same response as above.)

Round 2
Reviewer 1 Report
Comments and Suggestions for Authors
Authors answered to the pending questions and corrected the text accordingly. Only some typos were noted:
Line 95: Enterococcus species (4.1%, 36), streptococcus species (3.8%, 33), and proteus mirabilis (3.0%, 26) (data not shown)
Comment: First letter of Streptococcus spp and Proteus mirabils, should be capitalized
Author Response
All authors sincerely thank you for your kind comments of the manuscript.

Reviewer 2 Report
Comments and Suggestions for Authors
I appreciate the author's efforts in addressing my comments. I still have some remarks:
Uncomplicated cystitis is a leading cause of bacterial UTI; - cystis a form of UTI, not a cause.
Urinary tract infection (UTI) is among the most common bacterial infection in women affecting one million people worldwide each year, and is classified as uncomplicated or complicated based on the clinical presentation - not only based on the clinical presentation. there are additional factors that can contribute to this distinction ( imagistic, laboratory..)
row 54 - acute treatment?
row 55 - chronic recurrent UC - chronic or recurrent? Also, what is a chronic UC?
rows 95-96 - bacterial names misspelled
Thank you for this comment. We agree with your opinion. However, the present study is an internet-based, prospective, multicenter and national observational survey. So, we think that it would be appropriate to attach figures each for the comparisons of important results (e.g., ESBL positivity, fluoroquinolone resistance and 3rd generation cephalosporin resistance, respectively) by clinical factor in the present study. - the fact that is an internet-based, prospective, multicenter and national observational survey. has nothing to do with the figures. Moreover, the fact that the same parameter is analyzed (S/I/R or Pos/Neg), makes them suitable for grouping in only one figure.
row 154 - severe sequalae including frequent recurrences - recurrences are not severe, nor sequelae.
rows 153-154 - the exact phrase is already inserted in the introduction. Avoid duplication all over the manuscript, as this is not the only case.
We agree with your opinion and already mentioned the present study`s strengthness and possibility in discussion section as follows: the present study provides important information on the AMR status of E. coli in patients with UC in the Republic of Korea, and the continuous monitoring and accumulation of data enables the identification of changing AMR patterns over time and comparisons with other antimicrobial monitoring systems. In addition, our data can provide the evidence base for future antimicrobial treatment guidelines for UTIs. - I have read the limitations. All I said is that just by acknowledging them, does not make them dissapear. I suggested a way to overcome this issue.
Nationality, age, sex, outpatient or inpatient, association (community-acquired, healthcare-associated, hospital-acquired), uncomplicated or complicated, classification of UTI (cystitis, pyelonephritis, catheter-associated UTI, acute prostatitis, chronic prostatitis, urosepsis or asymptomatic bacteriuria), and recurrence or not were investigated. - all this terms should be defined as per the study's protocol.
rows 326-333, 336-341 should be completely rephrased, as they contain multiple, from an ID perspective. I suggest having at least these paragraphs double-checked by an Infectious diseases specialist/microbiologist.
microbial standards of each institute - further details should be provided.
The conclusion should focus more in the findings of this study. only one row and a half is not enough.
The reference list could still benefit from a expansion, considering that literature on this topic is more than extensive.
Best regards,
Comments on the Quality of English Languagemoderate
Author Response

(The authors gave the same response as above.)

Round 3
Reviewer 2 Report
Comments and Suggestions for Authors
I appreciate the author's responses. I just have some minor remarks, easily fixable:
Any patient aged ≥18 years woman - maybe something like "any female patient...." would be more appropriate.
ROWS 95-96 - "spp." should not be italicized.
If an ID specialist had checked the manuscript, maybe the text should have been double-checked by an English native person, as multiple errors (conjugation issues, typos, mischoose of words) were identified, misleading the reader.
It's much better with the figures this way
Best regards,